# Characterization of Plant-Derived Natural Inhibitors of Dipeptidyl Peptidase-4 as Potential Antidiabetic Agents: A Computational Study

**DOI:** 10.3390/pharmaceutics16040483

**Published:** 2024-04-01

**Authors:** Alomgir Hossain, Md Ekhtiar Rahman, Md Omar Faruqe, Ahmed Saif, Suzzada Suhi, Rashed Zaman, Abdurahman Hajinur Hirad, Mohammad Nurul Matin, Muhammad Fazle Rabbee, Kwang-Hyun Baek

**Affiliations:** 1Department of Genetic Engineering and Biotechnology, University of Rajshahi, Rajshahi 6205, Bangladesh; alamgir199817@gmail.com (A.H.); ekhtiarbdj@gmail.com (M.E.R.); rashedzaman@ru.ac.bd (R.Z.); nmatin@ru.ac.bd (M.N.M.); 2Department of Computer Science and Engineering, University of Rajshahi, Rajshahi 6205, Bangladesh; faruqe@ru.ac.bd; 3Department of Pharmacy, University of Rajshahi, Rajshahi 6205, Bangladesh; tamim.ahmedsaif@gmail.com; 4Department of Zoology, University of Rajshahi, Rajshahi 6205, Bangladesh; suzzada.ru@gmail.com; 5Department of Botany and Microbiology, College of Science, King Saud University, P.O. Box 2455, Riyadh 11451, Saudi Arabia; ahirad@ksu.edu.sa; 6Department of Biotechnology, Yeungnam University, Gyeongsan 38541, Gyeongsangbuk-do, Republic of Korea

**Keywords:** diabetes disease, dipeptidyl peptidase-4 (DPP-4), molecular docking, dynamic simulations, density functional theory, principal component analysis

## Abstract

Diabetes, characterized by elevated blood sugar levels, poses significant health and economic risks, correlating with complications like cardiovascular disease, kidney failure, and blindness. Dipeptidyl peptidase-4 (DPP-4), also referred to as T-cell activation antigen CD26 (EC 3.4.14.5.), plays a crucial role in glucose metabolism and immune function. Inhibiting DPP-4 was anticipated as a potential new therapy for diabetes. Therefore, identification of plant-based natural inhibitors of DPP-4 would help in eradicating diabetes worldwide. Here, for the identification of the potential natural inhibitors of DPP-4, we developed a phytochemicals library consisting of over 6000 phytochemicals detected in 81 medicinal plants that exhibited anti-diabetic potency. The library has been docked against the target proteins, where isorhamnetin, Benzyl 5-Amino-5-deoxy-2,3-O-isopropyl-alpha-D-mannofuranoside (DTXSID90724586), and 5-Oxo-7-[4-(trifluoromethyl) phenyl]-4H,6H,7H-[1,2]thiazolo[4,5-b]pyridine 3-carboxylic acid (CHEMBL3446108) showed binding affinities of −8.5, −8.3, and −8.3 kcal/mol, respectively. These compounds exhibiting strong interactions with DPP-4 active sites (Glu205, Glu206, Tyr547, Trp629, Ser630, Tyr662, His740) were identified. ADME/T and bioactivity predictions affirmed their pharmacological safety. Density functional theory calculations assessed stability and reactivity, while molecular dynamics simulations demonstrated persistent stability. Analyzing parameters like RMSD, RG, RMSF, SASA, H-bonds, MM-PBSA, and FEL confirmed stable protein–ligand compound formation. Principal component analysis provided structural variation insights. Our findings suggest that those compounds might be possible candidates for developing novel inhibitors targeting DPP-4 for treating diabetes.

## 1. Introduction

Diabetes is a long-term metabolic disorder characterized by elevated blood sugar levels (hyperglycemia) resulting from insufficient insulin production and/or inefficient insulin use [1]. It is a growing global health concern that affects millions of individuals worldwide. As of 2021, the International Diabetes Federation (IDF) estimated that 537 million adults globally, aged 20 to 79, have diabetes, representing approximately 10.5% of the global adult population [2]. The number of people with diabetes is expected to increase; projections indicate that by 2045, there may be 647 million people living with the disease [3]. Diabetes has significant health and economic implications, as it is linked to an increased risk of complications, including cardiovascular diseases [4], kidney failure [5], and blindness [6]. Furthermore, diabetes-related healthcare costs and productivity losses significantly burden healthcare systems and economies globally [7]. Efforts are being made to raise awareness, improve prevention strategies, and enhance diabetes management to mitigate the impact of this widespread condition.

There are various types of diabetic medicines available, each with a specific mechanism of action. The most common classes of diabetic medications include oral antidiabetic drugs such as metformin [8], sulfonylureas [9], DPP-4 inhibitors [10], and injectable medications such as insulin [11] and GLP-1 receptor [12]. DPP-4 inhibitors are the first class of oral antidiabetic drugs that were effectively designed as anti-hyperglycemic agents for diabetic treatment. These medicines work by specific mechanisms to lower blood sugar levels, either by elevating insulin secretion [13], increasing insulin sensitivity [14], reducing glucose production [15], or slowing down carbohydrate absorption in the intestines [16]. The selection of medication is influenced by several factors, such as the type and extent of diabetes, individual patient traits, and the existence of any complications [16]. Diabetic medicines play a crucial role in helping individuals with diabetes to achieve and maintain optimal blood glucose control, thereby reducing the risk of complications associated with the disease. However, they are not without limitations and potential adverse effects. One limitation is that these medications may not address the root cause of diabetes but instead focus on symptom management [17]. Additionally, some diabetic medications may have varying efficacy among individuals, requiring a trial-and-error approach to find the most suitable treatment [18]. Adverse effects can include hypoglycemia (low blood sugar) [19], gastrointestinal issues such as nausea [20] and diarrhea [21], weight gain [22], allergic reactions [23], and, in some cases, an increased risk of cardiovascular events [24]. Long-term use of certain diabetic medications, such as thiazolidinediones, may be associated with an elevated risk of bone fractures or bladder cancer [25].

Dipeptidyl peptidase-4 (DPP-4) is a serine aminopeptidase, commonly referred to as CD26, an enzyme that is necessary for regulating glucose metabolism [26] and immune function [27], and is one of the validated targets for diabetes therapy due to its regulatory effect on incretin hormones [28]. DPP-4 is expressed on the surface of many different cell types, including immune cells, endothelial cells, and epithelial cells [29]. DPP-4 possesses several crucial regions [30] responsible for catalytic and ligand binding sites: Asn85, Ser86, Arg147, Asn150, Val178, Trp187, Ile194, Glu205 and 206, Asn229, Thr231, Glu232, Asn281, Val289, Tyr547, Ser630, Tyr631, Val656, Tyr662 and 666, Asp708 and 709, Val711, and His740. Within the binding sites, there exist S1 and S2 primary pockets, and a Sub S1 or S3 sub-pocket [31]. Within the S1 pocket, either a substituted aromatic ring or a substituted saturated heterocycle is situated, while the P2 substituent occupies the S2 pocket and can bind either through covalent or non-covalent interactions. The two main interactions that cyanopyrrolidines have are as follows: the nitrile group forms reversible covalent bonds with the catalytically active serine hydroxyl group in the first interaction (Ser630) [32]. The second interaction is with the protonated amino group, which creates a network of hydrogen bonds with specific regions of the protein surface, namely Glu205 and 206, and Tyr662, which carry a negative charge. The S2 pocket is typically occupied by a ligand consisting of an aromatic heterocyclic ring or substituted fused ring [33]. The hydrophobic sub-S1 site is ideally filled with an aromatic group, and a connecting structure between the S1 and Sub-S1 binding elements is present. It also includes areas such as the S1-pocket Glu205, Glu206, Tyr547, Ser630, and His740, which contribute to ligand-binding active sites [34]. Appendix A shows that the DPP-4 protein has several crucial regions responsible for the catalytic and binding sites and their pockets S1 and S2, including residues.

The main function of DPP-4 is to cleave and deactivate incretin hormones, such as glucagon-like peptide-1 (GLP-1) [35] and glucose-dependent insulinotropic polypeptide (GIP) [36], which are involved in glucose homeostasis and insulin secretion [37]. By inhibiting the degradation of these incretin hormones, DPP-4 inhibitors can enhance their activity, leading to improved glucose control [38]. Beyond its metabolic role, DPP-4 is also involved in immune modulation [39], T-cell activation, and migration [40]. DPP-4 inhibitors have shown potential therapeutic benefits beyond glycemic control, including anti-inflammatory effects [41] and cardiovascular benefits [42]. Further research continues to elucidate the diverse roles of DPP-4 in physiological processes and its potential as a therapeutic target in conditions such as diabetes, inflammation, and cancer [43].

Plants have been utilized as medicines for thousands of years. Herb-based natural compounds are considered as novel, practical, and accessible agents in chronic diseases by inhibiting activation of causative enzymes [44]. Many plants have been used to treat diabetes, and as a source of medicines, interest in medicinal plants has increased [45,46]. Phytochemicals have shown promise as potential DPP-4 inhibitors for managing diabetes. Phytochemicals with DPP-4 inhibitory properties can modulate glucose metabolism and enhance insulin secretion [47], thus offering a natural approach to glycemic control. For instance, certain flavonoids, such as quercetin and apigenin, found abundantly in fruits, vegetables, and herbs, have demonstrated DPP-4 inhibitory activity in in vitro and in vivo studies [48,49,50]. These phytochemicals have the potential to lower blood glucose levels [51], improve insulin sensitivity [52], and reduce inflammation associated with diabetes [53]. The potential of natural sources as DPP-4 inhibitors for the development of novel compounds with therapeutic promise against diabetes have been extensively reviewed recently [54]. Incorporating phytochemical-rich foods or plant-based supplements into the diet may provide a complementary strategy to conventional diabetes management by harnessing the DPP-4 inhibitory properties of these natural compounds [55]. In silico studies can predict the inhibitory potential of phytochemicals, allowing for the selection of promising candidates for further investigation [56].

Current research in medicinal plant-based drug discovery involves intricate approaches using phytochemical, biological, and molecular techniques [57]. Computer-aided drug discovery approaches are implemented in boosting the drug-likeness, pharmacodynamics and pharmacokinetics of potential drug candidates [58,59]. These techniques include molecular design, docking simulations, and absorption, distribution, metabolism, excretion, and toxicity (ADMET) prediction [60,61,62]. A compound’s physicochemical properties provide information about its molecular interactions with biological systems, their impact on pharmacokinetics, and their influence on the compound’s overall drug-like attributes, such as adherence to rules like those of Lipinski, Ghose, and GSK. This study identified compounds that have the potential to bind to DPP-4. Additionally, these methods provided insights into the structural features and mechanisms underlying the interaction between phytochemicals and DPP-4, guiding the development of novel DPP-4 inhibitors derived from plant sources. This study may provide a valuable starting point; however, subsequent experiments are necessary to validate the efficacy and safety of phytochemicals as DPP-4 inhibitors for the treatment of diabetes.

## 2. Materials and Methods

### 2.1. Filtering Unwanted Sub-Substructures with Physiochemical Analysis

Filtering unwanted sub-substructures with physiochemical analysis is a process in virtual screening where undesired chemical components or molecular structures are removed based on their chemical properties, to reduce side effects and false positive results in drug discovery. Initially, 81 medicinal plants and their corresponding phytochemicals, identified by GC-MS and exhibiting anti-diabetic properties, were listed through an extensive literature review (Appendix A). RDKit version 2023.03 was used to assess compounds for pan-assay interference compounds (PAINS), Brenk properties, and NIH properties [63]. Common unwanted substructures included phenol-sulfonamides, rhodanines, phenol-esters, curcumin, enones, hydroxyphenyl hydrazones, catechols, toxoflavin, isothiazolones, analine, and quinones. To calculate various physicochemical and molecular properties of phytochemicals using RDKit v2023.03, various descriptors like molecular weight, rotatable bonds, hydrogen acceptor and donor, MLogP, TPSA, hetero atoms, molar refractivity, and aromatic rings were calculated. Physiochemical parameters were set to molecular weight: 200–480, mlogp: −0.4–4.15, hetero atoms: >1, MR: ≥40–130, and TPSA: ≤131.6. These criteria are often used as filters to eliminate compounds during the early stages of computational drug discovery.

### 2.2. Virtual Screening via Molecular Docking

Molecular docking predicts the binding interactions between ligands and target proteins using computational techniques, and thus helps to identify potential drug candidates and understand their binding mechanisms [64]. The PubChem database was used to download all of the filtered compounds in 3D SDF format. All ligand optimizations and energy minimization using mmff94 force field were utilized [65], and PyRx software v0.8 used the steepest descent optimization algorithm steps 2000 [66]. The RCSB Protein Data Bank was used to collect the DPP-4 receptor’s three-dimensional X-ray crystal structure (PDB ID: 4a5s) in pdb format. All heteroatoms and water molecules were removed, and hydrogen was added to generate the protein structure. Thereafter, the cleaned protein structure was optimized, and energy was minimized by utilizing the ff19SB force field [67] with the YASARA software, version 21.12.19 [68]. The molecular docking program AutoDock Vina v4.2.6 was used to carry out a molecular docking study. AutoDock Vina was used to convert PDB into PDBQT format for the input of proteins and ligands in the docking analysis [69]. The docking process took place in a 30 × 30 × 30 grid box and a center of x = 26.85, y = 12.60, z = 58.94, and dimensions of X: 51.35, Y: 66.93, and Z: 59.60 Å were kept. The docking outcomes were computed and arranged based on negative values, which indicated stronger binding affinities. To verify the docking study, the reference N7F inhibitor bonded with 4a5s was also subjected to docking against the target protein. All docking poses and 2D and 3D protein–ligand interactions were visualized and analyzed using Discovery Studio Visualizer v2.5.5 and UCSF Chimaera v1.17.3.

### 2.3. ADME/Tox and Bioactivity Analysis

Toxicological assessment of a compound at an early stage is crucial in the field of drug design and development. Traditionally, an in vivo animal model has been used to determine the toxicity of a compound. However, this approach is costly, time-consuming, and associated with ethical issues. Consequently, the computer-aided drug design process in silico toxicity test of chemical compounds can be considered useful. The ProTox-II server was utilized to predict the toxicity of the chosen compounds using the canonical simplified molecular-input line-entry system (SMILES) specification as the input syntax [70]. SwissADME [71] and pKCSM [72] (two publicly available web servers) were used to compute the compound’s absorption, distribution, metabolism, and excretion (ADME) properties. Both servers helped to determine and predict various pharmacokinetic and pharmacodynamic characteristics of the selected compound. The evaluation of the biological activities of three compounds was performed using the Molinspiration server. Here, a bioactivity score >0 indicated biologically active compounds, −5.0 to <0 denoted moderately active, and <−5.0 indicated biologically inactive compounds.

### 2.4. Density Functional Theory (DFT) Calculation

DFT is based on quantum mechanics and provides a highly accurate description of the distribution of electrons within a molecule, allowing for the calculation of various properties such as molecular energies, geometries, and electronic properties. The Gaussian 09 W software package [73] was used to calculate the various quantum mechanical properties. The electronic properties of the molecules were computed in their singlet ground state, without any charge and solvent, using the B3LYP (Becke, 3-parameter, Lee-Yang-Parr) [74] method within density functional theory and a 6–311 g(d,p) correlation function basis set. The DFT method was applied to assess the molecule’s reactivity through the investigation of different reactivity descriptors, which encompassed electron affinity, ionization potential, electronegativity (χ), electronic potential (μ), chemical hardness (η), chemical softness (ζ), and electrophilicity (ω) [75].

### 2.5. Molecular Dynamics Simulation

Molecular dynamics (MD) simulation is a computational approach that replicates the temporal movement and behavior of atoms and molecules, allowing for the simulation and analysis of their dynamic interactions within different systems. To understand the structural variations along the simulation trajectories, molecular dynamics simulations were conducted on the selected docked complexes [76]. The simulations were performed using YASARA [68] dynamics software (version 21.12.19), employing the AMBER14ffSB force field [67] for all calculations. Long-range electrostatic interactions were computed using the particle mesh Ewald method [77], while short-range van der Waals and Coulomb interactions were analyzed within an 8 Å cutoff radius [78]. During the simulations, a Langevin thermostat and a Monte Carlo barostat [79] were used to monitor the temperature and pressure, respectively. The total environment was set to a temperature of 298K, pH 7.4, 0.9% NaCl concentration, and water solvent system. To minimize the system, the steepest descent algorithm was used [80]. In the simulations, a time step of 1.25 fs [69] was utilized for a total duration of 100 ns, with snapshots taken at 100 ps intervals to analyze the trajectory data [77]. The parameters that were analyzed through the simulation trajectories included the number of hydrogen bonds, SASA, RG, RMSD, and RMSF [69,81,82]. Matplotlib Python library was used to visualize the plots generated from the MD simulations.

### 2.6. MM-PBSA Estimation and Free Energy Landscape Analysis

The results of MD simulations were utilized to measure the binding free energies of the compounds formed between proteins and ligands using the MM-PBSA technique [83]. This is a dependable and effective technique for simulating free energy and modeling molecular recognition, particularly in contexts like protein–ligand binding interactions [84]. Ligand binding energies were calculated using the md_analyzebindenergy.mcr of the YASARA Dynamics framework [85]. The foundation of this approach is the idea of breaking down the complex’s total free energy into its component elements, which include factors such as solvation, van der Waals, and electrostatic energies [83,86,87]. The binding free energy was estimated using the following equation:ΔG_bind_ = ΔG_complex_ − [ΔG_protein_ + ΔG_ligand_]

This equation can be further detailed as ΔG_bind_ = ΔG_MM_ + ΔG_PB_ + ΔG_SA-TΔS_.

In this context, ΔG_MM_ signifies the molecular mechanics interaction, which is the sum of electrostatic and van der Waals interactions, while ΔG_PB_ and ΔG_SA_ are polar and non-polar solvation energies, respectively, and T_ΔS_ is entropic contribution.

A free energy landscape (FEL) depicts the potential energy of a system about its pertinent variables [88], offering insights into the energy obstacles and favorable states a molecule explores during transitions between various conformations or states [89]. Based on the simulation trajectories, RMSD and RG values were computed for every frame, forming a density matrix that characterized a distinct area within the conformational space, and subsequently, the Gibbs free energy surface was determined by this density matrix and principles of statistical mechanics [90]. The 2D and 3D plot visualizations were created using Matplotlib v3.7.2.

### 2.7. Principal Component Analysis and Probability Density Function

Principal component analysis (PCA) in MD simulation [90] is a data reduction technique that identifies the essential collective motions and helps simplify complex molecular dynamics data by extracting the most significant structural fluctuations in the system [91]. Extracted atomic coordinates of the protein–ligand complex for each time step were used to calculate the covariance matrix. The diagonalization of the covariance matrix was performed to acquire eigenvectors and eigenvalues [92]. The eigenvectors were arranged in descending order according to their respective eigenvalues, and the coordinates in the reduced dimensional space were obtained by projecting the MD trajectory onto the principal components. The covariance construction matrix diagonalization and PCA were performed utilizing the scikit-learn v1.3 library. The probability density function (PDF) involves analyzing the distribution of structural properties over the trajectory to gain insights into the dynamic behavior of the complex [93]. The histogram values of RG and RMSD were normalized to represent a probability distribution. Gaussian kernel density estimation was calculated to estimate the PDF of the variables in a non-parametric way using scikit-learn v1.3, and arrays were stacked and sorted by implementing NumPy v1.25.2.

## 3. Results and Discussion

In many studies, DPP-4 inhibition has been revealed to decrease diabetes-induced beta cell dysfunction in in vitro and pre-clinical studies and has been related to beta cell mass and functional increases in several diabetic models [94]. Plants have been utilized as medicines for thousands of years. Inhibitors from medicinal plants have been used as complementary therapy in the treatment of several diseases like, cardiovascular diseases and diabetes mellitus. We have developed a library of phytochemicals and performed a series of experiments to detect potential therapeutic agents for diabetes mellitus.

### 3.1. Library Preparation and Filtration of Unwanted Substructures of Phytochemicals

Initially, 81 medicinal plants and their corresponding phytochemicals, identified by GC-MS, exhibiting anti-diabetic properties, were listed through an extensive literature review. RDKit version 2023.03 was used to assess compounds for pan-assay interference compounds (PAINS), Brenk properties, and NIH properties. Various descriptors like molecular weight, rotatable bonds, hydrogen acceptor and donor, MLogP, TPSA, hetero atoms, molar refractivity, and aromatic rings were calculated. Physiochemical parameters were set to molecular weight: 200–480, mlogp: −0.4–4.15, hetero atoms: >1, MR: ≥40–130, and TPSA: ≤131.6. We chose the compounds that fulfilled the criteria for further study. These criteria are often used as filters to eliminate compounds during the early stages of computational drug discovery.

Molecular filters help refine extensive chemical libraries to align with specific goals by eliminating unwanted chemical structures and properties. PAINS, BRENK, and NIH datasets are widely used [95] to detect and remove problematic chemical compounds, thus improving the effectiveness and reliability of drug discovery endeavors. We examined non-repetitive phytochemicals to identify unwanted substructures, applying criteria from the PAINS, BRENK, and NIH datasets. Among these, PAINS constituted 18.65%, BRENK accounted for 51.26%, and NIH contributed 6.59%. After applying this rigorous filtration process, the final set of phytochemicals consisted of only 23.5% of the total compounds. The molecules that were retained were more likely to fulfill the specified criteria and were thus suitable for further research in this study.

After screening the library, the best 10 compounds with high binding scores were selected (Table 1). Further, three compounds were studied due to their high docking scores and potential features.

### 3.2. Molecular Docking Analysis

Molecular docking was used to predict and analyze the interaction between filtered unwanted substructures of phytochemicals and DPP-4 protein at the molecular level. DPP-4 possesses a catalytic and ligand binding site. Simultaneously, the ligand’s primary amine establishes a hydrogen bond with Glu205 and Glu206 [33]. Docking volume defines the three-dimensional space within which molecular docking simulations are performed to predict the binding orientations and conformations of ligands within the binding pocket of a target protein. We performed site-specific docking, where the binding site of the target protein was subjected to docking against DPP-4. To evaluate the ligand positioning within the binding pockets, phytochemicals were subjected to docking against DPP-4 (Figure 1). Among the compounds, isorhamnetin, DTXSID90724586, and CHEMBL3446108 were selected due to their docking scores and interactions with DPP-4. As a benchmark, the native inhibitor N7F was selected, exhibiting a docking score of −8.1 kcal/mol (Table 2). Results demonstrated that isorhamnetin achieved a docking score of −8.5 kcal/mol and established four hydrogen bonds with residues Lys554, Asp545, Gly741, and Ser630 (Table 2). Additionally, it formed three hydrophobic bonds with residues Trp629, His748, and Tyr752, thereby interacting with DPP-4. The docking results analysis of isorhamnetin showed the presence of two bonds in the active site and three bonds in the ligand binding site and S1 pockets. These interactions indicate isorhamnetin’s capacity to disrupt crucial active sites. These bonds within the active sites probably play a substantial role in its strong attraction to protein. On the other hand, DTXSID90724586 exhibited a docking score of −8.3 kcal/mol against DPP-4, establishing three hydrogen bonds with residues Glu205, Glu206, and Tyr631 and five hydrophobic bonds with Tyr547, Val656, Val711, Tyr662, and Tyr666. Within the S1 pockets and ligand binding sites, it formed two hydrogen bonds and three hydrophobic bonds, underscoring its activity within these significant regions. Likewise, CHEMBL3446108 demonstrated a docking score of −8.3 kcal/mol against the target protein and established three hydrogen bonds with residues Arg125, Ser630, and Tyr631 (Table 2).

Figure 1 depicts the non-bonded interactions found in DPP-4′s active and catalytic sites. Additionally, two of the studied compounds formed three hydrophobic bonds with Val656, Tyr662, and Tyr666, thereby interacting with DPP-4. The interaction analysis showed the presence of one hydrogen and one hydrophobic bond in the S1 pockets. Analysis indicated these compounds form stable and strong binding with DPP-4 due to the numerous bonds established during the docking process. Specifically, these compounds form bonds with the catalytic and binding residues of the target molecule.

N7F establishes interactions with four hydrogen bonds involving residues Glu205, Val546, Trp629, and Ser630. Hydrogen bonds in docking facilitate initial recognition and binding, enhancing specificity. In simulations, their persistence signals stable protein–ligand complexes, while altered patterns may indicate changes in binding mode or stability. Additionally, N7F engages in four hydrophobic bonds with residues Lys554, Phe357, Tyr547, and Tyr666. Hydrophobic interactions in docking enhance binding affinity through nonpolar forces, aiding proper orientation. In simulations, persistent hydrophobic interactions signify stable complexes, while disruptions may hint at instability or binding pose changes. Isorhamnetin, classified as a flavonoid compound [96], exhibits a wide range of pharmacological effects, including cardiovascular and cerebrovascular protection, anti-tumor properties, anti-inflammatory activity, antioxidant effects, organ protection, and prevention of obesity [97]. It also involves the mechanisms of the control of PI3K/AKT/PKB, NF-κB, and MAPK [96], as well as other signaling pathways and the expression of related cytokines and kinases. Recent studies have reported that isorhamnetin displays pharmacological effects, including anti-bacterial and anti-virus. DTXSID90724586 has been demonstrated to exhibit various biochemical and physiological impacts. This compound has been shown to bind to protein through hydrogen bonding and hydrophobic interactions; as a result, it inhibits the activity of enzymes. It proved to possess anti-inflammatory, anti-cancer, and anti-viral properties and is involved in regulating gene expression and cell cycle progression. CHEMBL3446108 is an important carboxylic alpha-amino acid involved in biochemical and physiological processes.

This study considers DPP-4 as a promising therapeutic target for diabetes treatment. Many DPP-4 inhibitors, including sitagliptin, linagliptin, saxagliptin, alogliptin, vildagliptin, and anagliptin, have been developed; however, the pharmacokinetic considerations and adverse effects of these synthetic DPP-4 inhibitors remain a major concern. Therefore, our study demonstrated three phytocompounds screened from medicinal plants as potential natural DPP-4 inhibitors in the treatment of diabetes. Through molecular docking simulations, 1S-α-pinene, β-pinene, and dehydro-p-cymene from *Ocimum tenuiflorum* and Deazaxanthine have been discovered as potential inhibitors of the DPP-4 protein [98,99]. Yang et al. reported three phytocompounds, O-Ethyl-4-[(α-L-rhamnosyloxy)-benzyl] carbamate, isothiocyanate active ingredient and dipeptide as DPP-4 inhibitors from *Moringa oleifera* based on in silico pharmacokinetic analysis, which was the prime analysis to select the representative compounds for the treatment of chronic diseases such as diabetes and obesity [100].

### 3.3. ADME/Tox and Bioactivity Analysis

Properties such as physicochemical attributes, toxicity, selectivity, pharmacokinetics, and mode of action are also significant factors in determining whether a compound is a potential candidate for drug development. Some of these properties are summarized in Table 3. Drug-likeness properties make a molecule more likely to be a successful drug candidate [101]. This is closely associated with a compound’s physicochemical properties, which dictate the extent to which a compound interacts with biological systems and its behavior within the body [102]. ADME/T, drug-likeness, and bioactivity predictions were conducted for three of the examined compounds and N7F. The selected molecules satisfied Lipinski’s ‘Rule of Five’, indicating they are considered as lead compounds. Among the compounds, isorhamnetin, DTXSID90724586, CHEMBL3446108, and N7F exhibited superior gastrointestinal absorption, and CHEMBL3446108 and N7F were identified as substrates for p-glycoprotein, a drug efflux pump that actively removes compounds and hinders their accumulation. Both of these compounds demonstrated an inability to penetrate the blood–brain barrier. Additionally, DTXSID90724586 and N7F confirmed higher scores in synthetic accessibility compared to others. In toxicity parameters, isorhamnetin and CHEMBL3446108 displayed anticipated activity in immunotoxicity and hepatotoxicity, respectively, while showing inactivity in for mutagenicity, cytotoxicity, and carcinogenicity. Conversely, DTXSID90724586 and reference compound N7F shared the same inactive predictions across all aforementioned parameters. Both of these compounds were classified as low toxic, and DTXSID90724586 was assigned to toxicity class 6, whereas isorhamnetin, N7F, and CHEMBL3446108 received toxicity class 5, 5, and 4 designations, respectively (Table 4).

Molinspiration’s Bioavailability Suite uses proprietary methods, including chemical descriptors and machine learning/QSAR modeling, to predict bioactivity scores based on molecular structures. The trained model estimates the likelihood of a molecule exhibiting specific biological activity. The bioactivity scores of these compounds were predicted for assessing their activity as enzyme inhibitors and denoted as meaningfully active, moderately active, or inactive. Moderate activity was demonstrated by isorhamnetin in functions such as G protein-coupled receptor ligand, ion channel modulator, and inhibitor of proteases (Table 4). Nonetheless, it displayed notable biological efficacy as both a kinase inhibitor and a nuclear receptor ligand, implying possible therapeutic uses in autoimmune conditions, cancer treatment, and inflammatory disorders, as well as a function in controlling gene expression [103]. DTXSID90724586 exhibited moderate activity in roles such as kinase inhibitor and nuclear receptor ligand. However, it revealed biologically active efficacy as a GPCR ligand, ion channel modulator, and protease inhibitor, suggesting possible therapeutic application for biological conditions. In contrast, CHEMBL3446108 exhibited moderate activity across all mentioned biological parameters. This indicates that it has a moderate ability to participate in a wide range of biological functions across different protein families, affecting cellular ion flow, various physiological properties, and specific pathways. In comparison, N7F demonstrated significant activity as a GPCR ligand, kinase inhibitor, protease inhibitor, and enzyme inhibitor but was moderately active as an ion channel modulator and nuclear receptor ligand. As a result, these three highly promising inhibitors can influence a broad spectrum of biological and physiological processes via their interactions with biomolecules compared to the effects of N7F. DTXSID90724586 showed better results than all other compounds, and it outperformed all other biological receptors.

### 3.4. Frontier Molecular Orbital Analysis

FMO analysis is a computational technique used in quantum chemistry that involves calculating and examining the energies, shapes, and electron distributions of the highest occupied molecular orbital (HOMO) and lowest unoccupied molecular orbital (LUMO) in a molecule. HOMO-LUMO study was performed to describe the reactivity of molecular orbitals in organic molecules. The importance of the reactivity of the DPP-4 target was laid out in previous studies such as Shoombuatong and Watshara et al. Molecules with a small or no HOMO–LUMO are preferred for activity. This chemism is studied for potential inhibitors. FMO calculations assessed isorhamnetin, DTXSID90724586, CHEMBL3446108, and N7F ionization potential and electron affinity. Electronic descriptors included electronic properties such as E_HOMO_, E_LUMO_, ΔE_gap_, ionization potential, electron affinity, electronegativity, chemical potential, hardness, softness, electronic potential, and electrophilicity, which were calculated and are shown in Figure 2 and Table 5. In quantum chemistry, electronic energy (Eh) is a fundamental component used in computational methods to study and understand the behavior of molecules. The differences in Eh suggested variations in the stability and bonding interactions of these compounds. Higher energy values indicated more reactive compounds. Isorhamnetin had the lowest value, at −1143.775 Eh, while DTXSID90724586 had the highest energy value, at −1054.650 Eh. In contrast, N7F had the highest dipole moment, at 5.911 D, while DTXSID90724586 had the lowest value, at 1.544 D. CHEMBL3446108 and Isorhamnetin had intermediate dipole moment values of 3.697 D and 1.645 D, respectively. Meanwhile, differences in dipole moment values indicated variations in the charge distribution and polarity. For instance, N7F is highly polar, while DTXSID90724586 is relatively less polar.

The E_HOMO_ values ranged from −0.224 to −0.251 eV, with compound CHEMBL3446108 having the lowest value and isorhamnetin having the highest. The E_LUMO_ values varied from −0.014 to −0.092 eV, with DTXSID90724586 being the lowest and CHEMBL3446108 being the highest. The ΔE_gap_ ranged from −0.15 to −0.222 eV, where isorhamnetin had the widest energy gap. Ionization potential values varied between 0.224 eV and 0.251 eV. Electronegativity values ranged from 0.507 to 0.546 eV, with the highest value corresponding to CHEMBL3446108. The chemical potential spanned from 0.125 to 0.343 eV. Hardness values were in the range of 0.075 to 0.111 eV, with DTXSID90724586 having the highest hardness. Softness values ranged from 4.504 to 6.667 eV^−1^, and electronic potential values fluctuated between −0.125 and −0.172 eV. Electrophilicity varied between 0.070 and 0.744 eV.

The lead compounds, as reflected in their electronic descriptors, possessed different electronic structures compared to the control. Specifically, isorhamnetin had a lower E_HOMO_ and a narrower energy gap, indicating potential differences in electronic properties. N7F exhibited a slightly lower E_HOMO_ but a comparable energy gap, while CHEMBL3446108 demonstrated a higher E_LUMO_ and ionization potential, suggesting variations in ionization behavior. Moreover, all three compounds diverged from the control in terms of electronegativity and electrophilicity, indicating differences in their chemical reactivity and potential interactions with the reference molecule (N7F).

### 3.5. Molecular Dynamics Simulation

Post-dock analysis relies on molecular dynamics (MD) simulations to examine the time-dependent stability and atomic motions of biological compounds, which are essential. MD simulation was conducted for 100 ns to gain insights into the structural behavior, binding interactions, and flexibility of DPP-4, the DPP-4–isorhamnetin complex, DPP-4–DTXSID90724586 complex, and DPP-4–CHEMBL3446108 complex. The findings yielded six distinct conclusions, which encompassed parameters like Cα atom RMSD, RMSF, RG, SASA, the presence of intermolecular hydrogen bonds, and MM-PBSA (Figure 3). The RMSD trajectories of the DDP-4–isorhamnetin complex showed an initial jump of relatively stable fluctuations between 39 and 57 ns of 2.20 to 2.74 Å, followed by decreased values in its trajectory. Following that, the complex exhibited variations, with trajectory values reaching a maximum of 2.74 Å and averaging 1.69 Å. Likewise, the DPP-4–DTXSID90724586 complex showed stability at first but after 54 to 55 ns, showed minor fluctuations; after that, it also reached a stable state. The RMSD value for this complex was a maximum of 2.43 Å, with an average of 1.58 Å. The DPP-4–CHEMBL3446108 complex exhibited an initial jump reaching 2.417 Å at 18 ns, and after that, showed several stabilities. Nevertheless, at 82 ns, it again showed high fluctuation. Afterward, the complex displayed fluctuations, with a maximum trajectory value at 2.65 Å and an average of 1.85 Å. Conversely, the reference complex, N7F, demonstrated significant fluctuations between 43 to 92 ns, with its RMSD trajectory reaching a peak value of 2.46 Å and averaging at 1.76 Å. Among these complexes, the RMSD trajectories indicate that all compounds retained stable configurations throughout the simulation duration, showing no substantial structural alterations from their initial states [104]. Moreover, the latter two compounds demonstrated enhanced stability in comparison to N7F.

In the RMSF trajectories of the interactions involving the four compounds, there were similar regions of flexibility within amino acid residues 95–97, 237–253, 278–281, 332–336, and 676–679 (Figure 3). To be precise, the DPP-4–isorhamnetin, DPP-4–DTXSID90724586, DPP-4–CHEMBL3446108, and DPP-4–N7F complexes exhibited average RMSF values of 1.16 Å, 1.17 Å, 1.22 Å, and 1.16 Å, respectively. In the isorhamnetin binding site, residues such as Asp545, Lys554, Ser630, Gly741, Trp629, His748, and Tyr752 are involved. The DTXSID90724586 compound binds to residues including Glu205, Glu206, Tyr631, Tyr547, Val656, Val711, Tyr662, and Tyr666. Meanwhile, the binding site for the CHEMBL3446108 compound involves residues Arg125, Tyr631, Ser630, Tyr666, Val656, and Tyr662. The RMSF values of specific regions within the complex indicate their equilibrium states. Among these compounds, Arg125 residue showed the highest RMSF value of 1.26 Å. This suggests that all protein complexes showed limited flexibility in specific areas, emphasizing the stable structural basis of the protein. The RG trajectories of the DPP-4–isorhamnetin, DPP-4–DTXSID90724586, and DPP-4–N7F complexes exhibited comparably similar patterns up to 27 ns. However, beyond that point, the latter complex displayed relatively elevated trajectories compared to the former two. On the other hand, the DPP-4–CHEMBL3446108 complex showed higher fluctuation than the other complexes during the simulations. The DPP-4–isorhamnetin complex reached its highest RG value at 83 ns, of 27.39 Å, while the DPP-4–DTXSID90724586, DPP-4–CHEMBL3446108, and DPP-4–N7F complexes reached 27.36 Å, 27.62 Å, and 27.27 Å at 66 ns, 80 ns, and 77 ns, respectively. Notably, the DPP-4–CHEMBL3446108 complex RG value showed more fluctuation than the others during the simulation. Nevertheless, its subsequent behavior conformed to the patterns observed in the other three complexes, ultimately reaching stability over time. The average RG values for the four complexes were computed to be 27.19 Å, 27.17 Å, 27.28 Å, and 27.06 Å, respectively. As a result, the structural tightness and the folding attributes of the protein–ligand interactions remained consistent throughout the simulation.

SASA indicates how much area of a molecule’s surface is available to the solvent molecules. The SASA trajectories of the DPP-4–isorhamnetin complex showed little fluctuation initially at 4 ns, then remained stable until 57 ns, but in the end, it gained stability. Almost the same behavior was shown in the other two complexes, DPP-4–DTXSID90724586 and DPP-4–N7F. During the simulation time, the DPP-4–CHEMBL3446108 complex showed fluctuating trajectories. It demonstrated that this complex had a significant degree and a much higher increase in surface area expansion of the protein. The other three complexes showed relatively moderate protein surface area expansion. Hydrogen bond trajectories of the four complexes remained stable during the simulation process, indicating the rigidity of each complex and low fluctuation, as indicated in RMSF trajectories except for some regions, which were aligned with other descriptor results. The analysis of hydrogen bonds in the intermolecular binding revealed that DPP-4 only and DPP-4–DTXSID90724586, DPP-4–CHEMBL3446108, and DPP-4–N7F complexes exhibited a similar number of hydrogen bonds within their molecules. However, the DPP-4–isorhamnetin complex displayed a higher number of hydrogen bonds than these four, indicating its stronger stability. Analysis of these various results revealed that, despite having slight conformational changes during MD simulation, three of these complexes had stability similar to that of the N7F reference complex.

### 3.6. MM-PBSA Estimation and Free Energy Landscape Analysis

The MM-PBSA determined and analyzes the protein–ligand interaction and binding energies of these biomolecular complexes [105]. MD simulations conducted in YASARA were employed to calculate ligand binding energies (Figure 3). In this context, higher positive values indicate a stronger binding affinity, whereas negative values do not necessarily indicate a lack of binding [106]. It is crucial to emphasize that YASARA’s calculations for binding energy are not appropriate for making absolute energy predictions because they do not account for entropy factors [107]. Therefore, the energies are assessed qualitatively to recognize noticeable patterns. The MM-PBSA binding energies of the DPP-4–isorhamnetin, DPP-4–DTXSID90724586, DPP-4–CHEMBL3446108, and DPP-4–N7F complexes were computed as −35.60 kcal/mol, 48.58 kcal/mol, 53.62 kcal/mol, and 37.57 kcal/mol, respectively. The reported binding energies were the thermal MM-PBSA. The trajectories are given in Figure 3. Computed values of −35.60 kcal/mol, 48.58 kcal/mol, 53.62 kcal/mol, and 37.57 kcal/mol were the average from 0 ns to 100 ns simulation runtime. It was indicated by their robust binding to the protein. The results from the MD simulation showed that these complexes remained stable and structurally rigid in their bound states, especially when compared to the reference N7F compound. The molecular docking was validated through MM-PBSA analysis of MD simulation. MM-PBSA analysis complements molecular docking by providing a more detailed and accurate estimation of binding free energy and affinity, incorporating solvation effects, handling conformational flexibility, and validating the results obtained from docking simulations.

The free energy landscape (FEL) reflects the complex, multidimensional connections among conformational states, structural stability, and thermodynamic favorability [89]. It helped identify the states with the lowest energy, transition paths, and the overall dynamic behavior of the complexes. Areas with lower RG and RMSD values indicated more compact and structurally stable configurations (Figure 4). GFE (Gibbs free energy) of a protein–ligand complex is computed by decomposing the total free energy into contributions from molecular mechanics energy and solvation energy. When these areas also had reduced GFE, it indicated that they represented thermodynamically favorable states. On the other hand, higher RMSD values indicated structural variations, which might align with less stable configurations. Increased GFE in these regions implied the presence of higher energy barriers and less favorable states [108]. The energy minima, visible as valleys, indicated the most stable configurations. These parameters are observed from the FEL plot. These results were also reported by Shaikh and Sibhghatulla et al. in their previous study. Transition states, marked by an increase in RMSD and RG values, suggested possible structural alterations. The steepness of the FEL surface represented the energy barriers for moving between these states.

### 3.7. Principal Component Analysis and Probability Density Function

To assess and identify correlated movements and dynamic regions within the protein, fluctuations, changes in atomic positions, and flexibility were examined using a PCA score plot [109]. By employing two eigenvectors derived from PCA analysis conducted over a 100 ns simulation, the two-dimensional projections and covariance matrix were calculated for these complexes. In this representation, the color shifted from blue and white to red as the simulation progressed. The 2D patterns illustrating protein motions at critical points, such as when the ligand binds, were closely examined. The impact of ligands on the protein and ligand interactions, and on the conformational dynamics of DPP-4, was examined. This examination revealed interactions with important amino acid residues. The proportions of eigenvector contributions to the total conformational variance were determined to be 45.32% for isorhamnetin, 23.35% for DTXSID90724586, 24.66% for CHEMBL3446108, and 28.26% for N7F, respectively (Figure 5). Each eigenvector represented a principal component of conformational mode, and its contribution to the total variance indicated the importance of that mode in describing the overall dynamics of the system. These values suggested that there were only slight coordinate shifts, indicating stability during the simulation. Lower PC values were linked to a decreased ability to bind, as shown by the red color, and this corresponded to the RG trajectories, which indicated restricted mobility and structural changes in the DPP-4 complexes. Significant variations in the conformation and compactness of the protein structure could impede interactions with substrates.

The analysis of the PDF reveals trends, patterns, and possible structural information regarding the dynamic behaviors of the complexes [110]. The PDF provides information about the distribution of structural states of a biomolecular system. It indicates the probability density of observing specific combinations of radius of gyration (RG) and root-mean-square deviation (RMSD) values within the system. The color bar represents the intensity or probability density of observing specific combinations of RG and RMSD values. A good PDF represented a well-defined, smooth distribution with clear peaks of high density, covering the relevant range of RG and RMSD. The DPP-4–isorhamnetin, DPP-4–DTXSID90724586, DPP-4–CHEMBL3446108, and DPP-4–N7F complexes displayed elevated PDF values in specific regions, namely 27.3-2.5, 27.4-2.5, 27.3-2.0, and 27.1-2.5 in the RG-RMSD space (Figure 6). These findings suggested that these particular combinations of compactness and structural deviation are more likely to occur. On the other hand, regions with lower probability density indicated less persistent structural forms. When the contour lines were closer together, it implied that RG and RMSD had a stronger correlation or underwent coordinated changes. In contrast, wide space contour lines indicated more noticeable transitions or a weaker direct link between RG and RMSD.

In summary, this study was performed to predict a novel potential phytochemical-based inhibitor, i.e., isorhamnetin, DTXSID90724586, and CHEMBL3446108, against DPP-4 via molecular docking, a deep learning model, density functional theory analysis, and molecular dynamics simulation. Furthermore, the research investigated the pharmaceutical attributes, MM-PBSA, free energy landscape, PDF, PCA analysis, and various biological aspects of these compounds to evaluate their potential as drugs targeting DPP-4. Our findings might help to develop potential drug candidates to fight abnormalities and diseases associated with DPP-4.

## 4. Conclusions

The present research exhibited that selected phytochemicals from medicinal plants and having anti-diabetic properties bind with DPP-4 with different poses. Isorhamnetin, DTXSID90724586, and CHEMBL3446108 were selected based on their binding affinity. AMDET and bioactive potential prediction indicated their drug-like qualities without toxicity concerns. Moreover, the utilization of DFT calculations, which included an examination of the HUMO-LUMO levels, provided insights into the reactivity and sustainability of the identified compounds. A comprehensive understanding of the protein–ligand interaction mechanisms was achieved through molecular dynamics simulation and principal component analysis; it verified the stability of the protein–ligand complexes, as no significant conformational changes occurred. The present in silico results reinforce the approach of developing potential DPP-4 inhibitors from natural resources. To ensure the DPP-4 inhibitors’ success, further validation through in vivo and in vitro experiments is required.

## Figures and Tables

**Figure 1 pharmaceutics-16-00483-f001:**
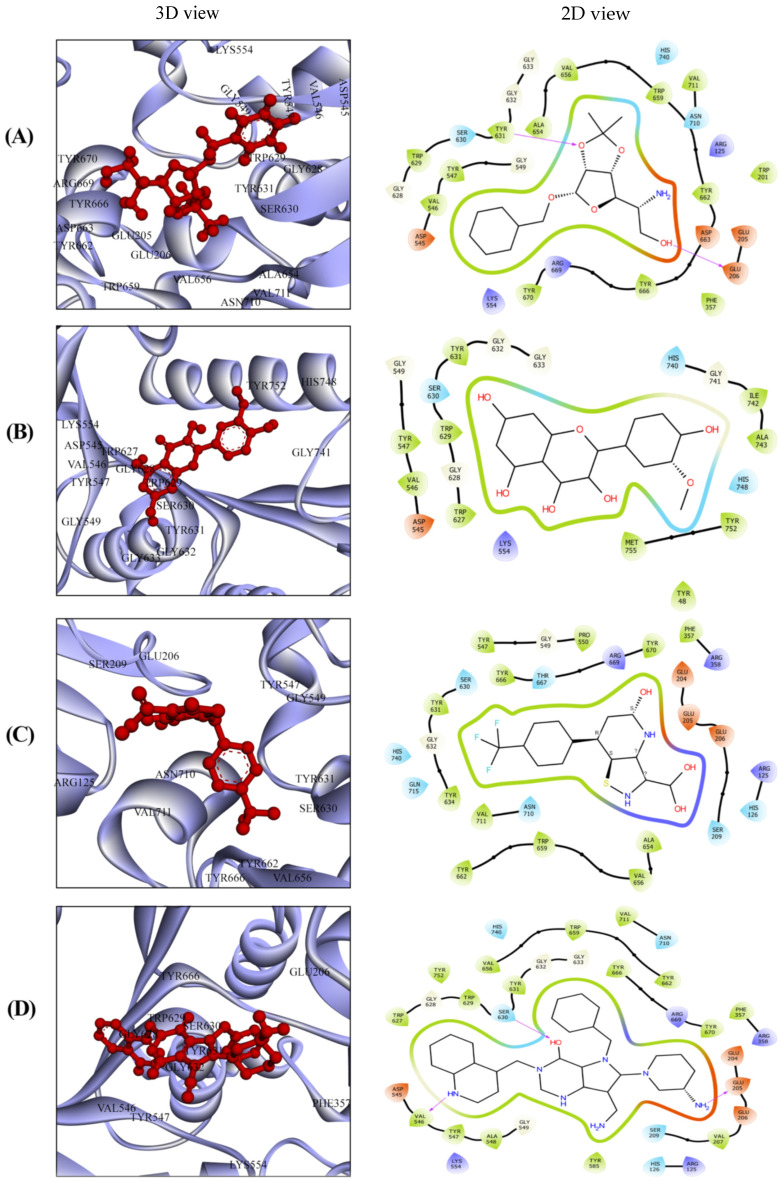
Molecular docking configurations and interactions with target protein. The 3D and 2D binding modes for the selected compounds within the active and catalytic sites of the complexes. (**A**) Isorhamnetin, (**B**) DTXSID90724586, (**C**) CHEMBL3446108, (**D**) N7F.

**Figure 2 pharmaceutics-16-00483-f002:**
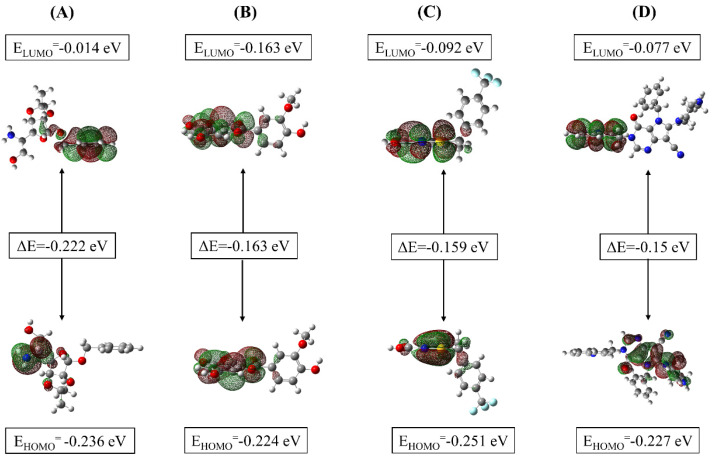
The ground state molecular orbital distribution plots of (**A**) isorhamnetin, (**B**) DTXSID90724586, (**C**) CHEMBL3446108, and (**D**) N7F at the DFT/SDD level of theory in the gas phase.

**Figure 3 pharmaceutics-16-00483-f003:**
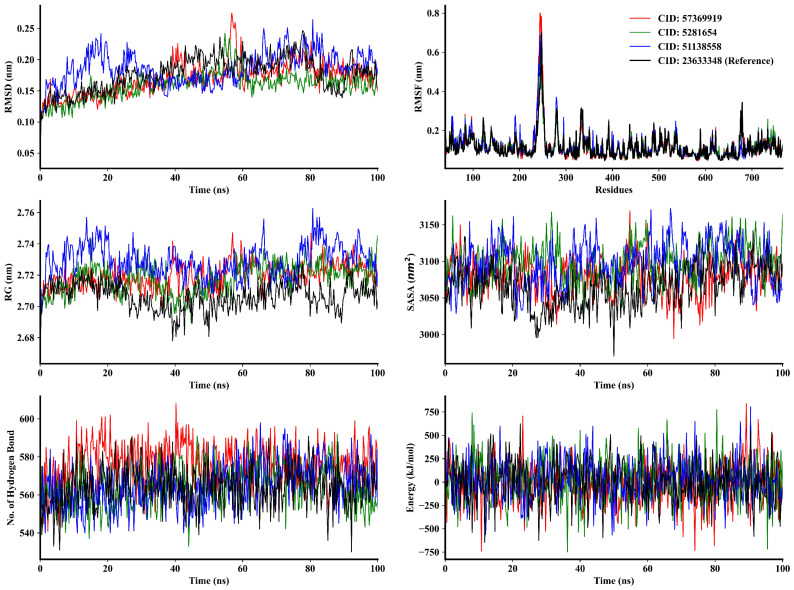
The structural characteristics of isorhamnetin (CID: 5281654), DTXSID90724586 (CID: 57369919), CHEMBL3446108 (CID: 51138558), and N7F (CID: 23633348) complexed with DPP-4 were investigated through 100 ns molecular dynamics (MD) simulations for each system. Various analyses, including RMSD for an alpha carbon atom, RMSF for assessing amino acid flexibility, RG for examining compactness and rigidity, SASA for evaluating volume changes of protein, hydrogen bond counts, and MM-PBSA for studying interaction energies in biomolecular complexes, were performed on the MD trajectories. The resulting MD plots were visualized using Matplotlib version 3.7.

**Figure 4 pharmaceutics-16-00483-f004:**
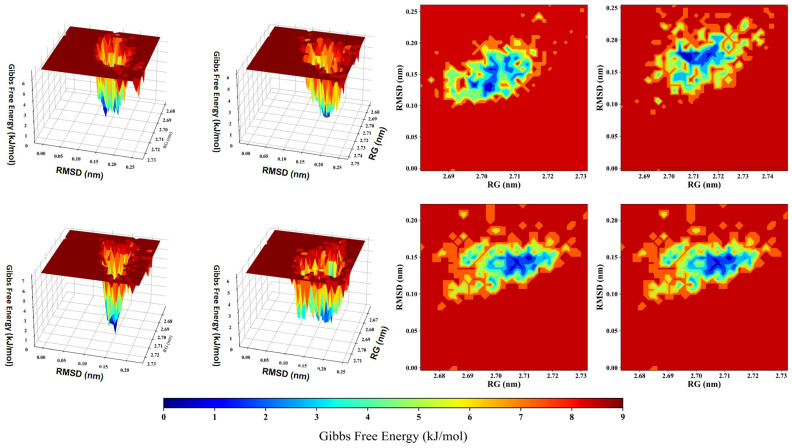
Three-dimensional (3D) and two-dimensional (2D) free energy landscape (FEL) depictions of protein–ligand complexes are accompanied by RMSD and RG structural parameters, offering insights into preferred energy states and ligand binding modes.

**Figure 5 pharmaceutics-16-00483-f005:**
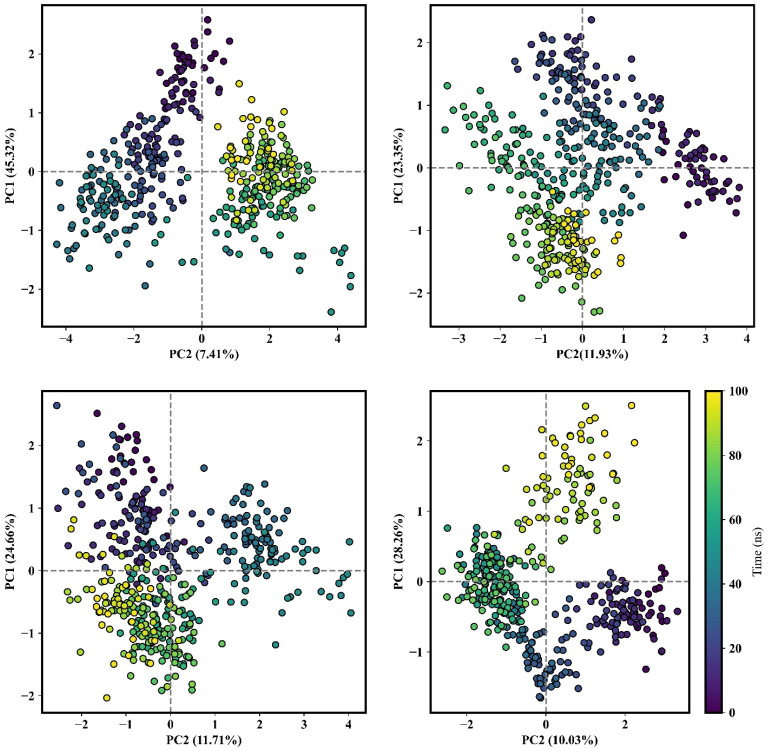
Principal component analysis (PCA) on MD simulations was performed for the DPP-4–isorhamnetin, DPP-4–DTXSID90724586, DPP-4–CHEMBL3446108, and DPP-4–N7F complexes. Each dot on the x and y axes represents a complex’s conformation. The spread of purple and yellow dots indicates the extent of conformational changes during the simulation, with purple signifying the start, teal representing intermediate stages, and yellow signifying the end of the simulation.

**Figure 6 pharmaceutics-16-00483-f006:**
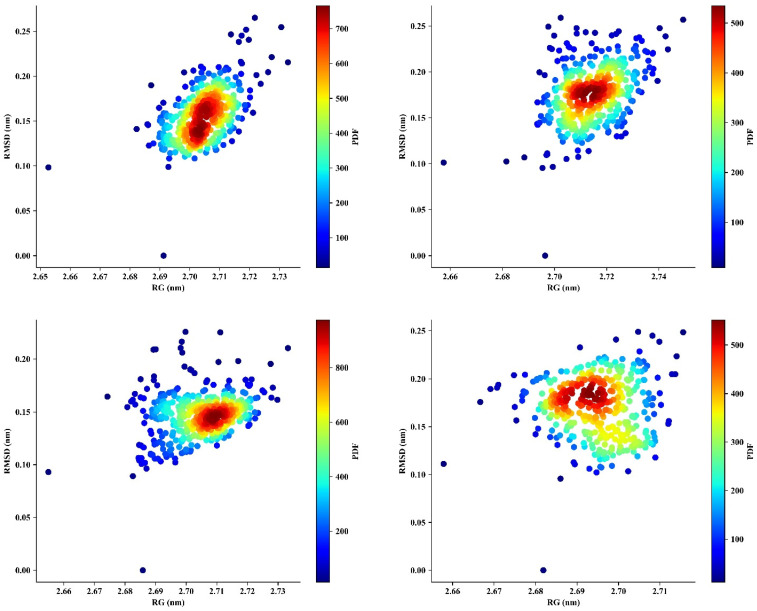
Probability density function (PDF) plots were created to show the distribution of RMSD and RG values for four protein–ligand complexes: DPP-4–isorhamnetin, DPP-4–DTXSID90724586, DPP-4–CHEMBL3446108, and DPP–4-N7F. These PDF analyses offer information about the complexes’ conformational dynamics and structural stability by illustrating the range and probability of various RMSD and RG values.

**Table 1 pharmaceutics-16-00483-t001:** Binding scores of the best 10 compounds.

No.	Compound CID	Compound Name	Binding Score
1	5281654	Isorhamnetin	−8.5
2	57369919	Benzyl 5-Amino-5-deoxy-2,3-O-isopropyl-alpha-D-mannofuranoside	−8.3
3	51138558	5-Oxo-7-[4-(trifluoro-methyl)phenyl]-4H,6H,7H-[1,2]thia-zolo[4,5-b]pyridine 3-carboxylic acid	−8.3
4	520919	2-(3-Methoxyphenyl)-5-phenyloxazole	−8.2
5	10680	Flavone	−8.2
6	72307	Sesamin	−8.0
7	243761	16alpha-Hydroxyprogesterone	−8.0
8	550072	Dasycarpidan-1-methanol, acetate (ester)	−7.9
9	631428	8-[1-Adamantyl]-1,3-diamino-5,6-dihydrobenzo[f]quinazoline	−7.9
10	5281607	Chrysin	−7.9

**Table 2 pharmaceutics-16-00483-t002:** Docking scores and non-bond interaction between DPP-4 and three selected potential compounds, with an asterisk (*) indicating the reference compound.

No.	Compound Name	PubChem CID	Binding Affinity (kcal/mol)	Residue in Contact	Interaction Type	Distance (Å)
1	Isorhamnetin	5281654	−8.5	ASP545	H-bond	2.924
				LYS554	H-bond	2.630
				SER630	H-bond	2.856
				GLY741	H-bond	2.469
				TRP629	Pi-pi stacked	3.741
				HIS748	Pi-alkyl	4.881
				TYR752	Pi-alkyl	5.489
2	Benzyl 5-Amino-5-deoxy-2,3-O-isopropyl-alpha-D-mannofuranoside (DTXSID90724586),	57369919	−8.3	GLU205	H-bond	2.901
				GLU206	H-bond	2.471
				TYR631	H-bond	2.392
				TYR547	Pi-pi stacked	3.773
				VAL656	Alkyl	4.888
				VAL711	Alkyl	4.614
				TYR662	Pi-alkyl	4.610
				TYR666	Pi-alkyl	4.003
3	5-Oxo-7-[4-(trifluoro-methyl) phenyl]-4H,6H,7H-[1,2]thia-zolo[4,5-b]pyridine 3-carboxylic acid (CHEMBL3446108)	51138558	−8.3	ARG125	H-bond	2.885
				TYR631	H-bond	2.856
				SER630	H-bond	3.584
				TYR666	Pi-pi stacked	4.030
				VAL656	Alkyl	4.749
				TYR662	Pi-alkyl	4.035
4	N7F *	23633348	−8.1	GLU205	H-bond	2.306
				VAL546	H-bond	2.048
				SER630	H-bond	2.289
				TRP629	H-bond	2.256
				LYS554	Pi-cation	4.782
				TYR666	Pi-pi stacked	4.611
				PHE357	Pi-alkyl	4.761
				TYR547	Pi-alkyl	5.188

N7F: native inhibitor (reference compound).

**Table 3 pharmaceutics-16-00483-t003:** Physicochemical and pharmacological profiles of the main three potential candidates and reference compounds derived using RDKit.

	Isorhamnetin	DTXSID90724586	CHEMBL3446108	N7F
**Physiochemical parameters**				
MolWt	316.05	309.15	342.02	489.57
NumHAcceptors	7	6	4	5
NumHDonors	4	2	2	1
MolLogP	2.291	0.767	3.332	1.674
NumRotatableBonds	2	5	2	5
TPSA	120.36	83.17	79.29	105.76
MolMR	81.131	78.778	75.605	147.66
NumHeteroatoms	7	6	9	7
NumAromaticRings	3	1	2	3
**Pharmacological parameters**				
PAINs	No	No	No	No
Brenk	No	No	No	No
NIH	No	No	No	No
Lipinski	Yes	Yes	Yes	Yes
Ghose	Yes	Yes	No	No
GSK	Yes	Yes	No	No

**Table 4 pharmaceutics-16-00483-t004:** Pharmacokinetics, toxicity, and bioactivity assessment of the hit compounds, i.e., isorhamnetin, DTXSID90724586, CHEMBL3446108, and N7F.

		Isorhamnetin	DTXSID90724586	CHEMBL3446108	N7F
**ADME**	GI absorption	High	High	High	High
	BBB permeant	No	No	No	No
	P-GP substrate	No	No	Yes	Yes
	Synthetic	3.26	4.60	3.54	4.30
	Log P *o*/*w* (Cons)	1.65	1.00	2.68	3.07
	Log S (ESOL)	Soluble	Soluble	Soluble	Moderately soluble
**Toxicity**	Carcinogenicity	Inactive	Inactive	Inactive	Inactive
	Hepatotoxicity	Inactive	Inactive	Active	Inactive
	Immunotoxicity	Active	Inactive	Inactive	Inactive
	Mutagenicity	Inactive	Inactive	Inactive	Inactive
	Cytotoxicity	Inactive	Inactive	Inactive	Inactive
	Toxicity class	5	6	4	5
**Bioactivity**	GPCR ligand	−0.10	0.29	−0.06	0.29
	Ion channel modulator	−0.26	0.06	−0.11	−0.10
	Kinase inhibitor	0.25	−0.01	−0.45	0.13
	Nuclear receptor ligand	0.28	−0.05	−0.17	−0.23
	Protease inhibitor	−0.30	0.43	−0.23	0.14
	Enzyme inhibitor	0.22	0.78	−0.09	0.32

**Table 5 pharmaceutics-16-00483-t005:** Global reactivity descriptors (eV) for four selected potential compounds were determined using the DFT B3LYP/3–21 g* basis set approach. These electronic characteristics were derived from the HOMO and LUMO energy levels.

Properties	Isorhamnetin	DTXSID90724586	CHEMBL3446108	N7F
Electronic energy (Eh)	−1143.775	−1580.202	−1572.160	−1054.650
Dipole moment (D)	1.645	5.911	3.697	1.544
E_HOMO_ (eV)	−0.224	−0.227	−0.251	−0.236
E_LUMO_ (eV)	−0.061	−0.077	−0.092	−0.014
ΔE_gap_ (eV)	−0.163	−0.15	−0.159	−0.222
Ionization potential (eV)	0.224	0.227	0.251	0.236
Electron affinity (eV)	0.061	0.077	0.092	0.014
Electronegativity (eV)	0.530	0.538	0.546	0.507
Chemical potential (eV)	0.142	0.152	0.343	0.125
Hardness (eV)	0.081	0.075	0.079	0.111
Softness (eV^−1^)	6.172	6.667	6.329	4.504
Electronic potential (eV)	−0.142	−0.152	−0.172	−0.125
Electrophilicity (eV)	0.124	0.154	0.744	0.070

## Data Availability

Data are within the article and Appendix A.

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
