# Peer review of "Characterization of Plant-Derived Natural Inhibitors of Dipeptidyl Peptidase-4 as Potential Antidiabetic Agents: A Computational Study"

_pharmaceutics, 2024, doi:10.3390/pharmaceutics16040483_

Round 1

Reviewer 1 Report

Comments and Suggestions for Authors

The authors report the identification and in silico characterization of potential DPP4 inhibitors from a medicinal plants database. The manuscript will be of interest to those working in antidiabetic drug design and in computer aided drug design. However, there are some points that deserve attention.

1.- A lot of information in results and discussion section must be moved to introduction section because is more a theoretical support than analysis or discussion of the data. For example, lines 143-179.

2.- Please include a Table with the binding score of the best ten compounds and discuss the reasons to select the three compounds studied.

3.- Please refer to the compounds studied in the same manner along the text, because sometimes their CID code is used instead their name.

4.- A deeper discussion of the data is missing, in its actual form, it is more a description of the results than an analysis related with information in literature.

5.- A general comment, in its actual form the scientific soundness of the work is not clear, it is an interesting approach but it is important to highlight its contribution to knowledge.

Comments on the Quality of English Language

Minor editing of English language required.

Author Response

Dear Editors and Reviewers: We would like to thank you for your time and effort in reviewing our manuscript, and valuable suggestions and corrections to improve the quality of our manuscript. We have carefully addressed all of the issues raised in the comments and incorporated in the text. The following is a point-by-point response to the reviewer's comments. We have marked all revised items in red color in the text according to the editors’ suggestions.

General Comments:

The authors report the identification and in silico characterization of potential DPP4 inhibitors from a medicinal plants database. The manuscript will be of interest to those working in antidiabetic drug design and in computer aided drug design. However, there are some points that deserve attention.

Author Response: We thank the reviewer for careful reading of the manuscript, critical comments, and useful suggestions to improve our manuscript. Hopefully, the reviewers will value our efforts behind this manuscript.

Other comments

Q1.- A lot of information in results and discussion section must be moved to introduction section because is more a theoretical support than analysis or discussion of the data. For example, lines 143-179.

Author Response: Thank you for the professional comments. We have substantially improved the introduction as well as the results and discussion section following your instructions. Most of the general discussion has been moved from the results to the introduction section.

Q2.- Please include a Table with the binding score of the best ten compounds and discuss the reasons to select the three compounds studied.

Author Response: We have included one table with the binding score of the best ten compounds. Further, we have discussed the reasons for selecting the three compounds studied.

Q3.- Please refer to the compounds studied in the same manner along the text, because sometimes their CID code is used instead their name.

Author Response: We have carefully checked the text thoroughly and corrected the compound name in the same manner.

Q4.- A deeper discussion of the data is missing, in its actual form, it is more a description of the results than an analysis related with information in literature.

Author Response: We appreciate this important comment. We have improved the discussions based on the results obtained and cited various results related to information in the literature.

Q5.- A general comment, in its actual form the scientific soundness of the work is not clear, it is an interesting approach but it is important to highlight its contribution to knowledge.

Author Response: Thank you very much for the professional comments. We have tried to substantially improve the manuscript. We hope now it is clear for its scientific soundness. 

Reviewer 2 Report

Comments and Suggestions for Authors

Manuscript “pharmaceutics-2889854” entitled “Characterization of Plant Derived Natural Inhibitors for Dipeptidyl Peptidase-4 as Potential Antidiabetic Agents: A Computational Study”  describes an in silico drug design study on DPP-4 target but lacks critical data and conclusions for the reader. I recommend major revision to amend the text:

Abstract: Line 32-44 – This info is not suitable for the abstract. Keep the abstract short and concise to include results only.

Line 110-111; please provide the reference.

Line 115-116; “Using computational approaches, this study 115 identified phytochemicals that exhibit favorable interactions and binding affinities with 116 the active site of DPP-4.” This study identified the componds that have potential to bind to DPP-4. Rephrase!

Line 132: For example Lipinski is not a filter to access drug-likeness. Each filter You describe has a specific purpose. Do You think phytochemical adhere to all Your employed filters in general, is it sensible to apply Lipinski filter at the beginning phytochemical design stage ?

Line 138: “After applying this rigorous filtration process, the 138 final set of phytochemicals consisted of only 23.5% of” 23 % of What? Please include the chapter: Library preparation where You describe in detail how You prepared the libraray and how many compounds are in there. Then You describe which compounds are filtered off in each step and supply the complete library in the supp info.

Line 143-165: Prepare a figure where you indicate these sites on the target and display your poses in this context so the reader can evaluate the docked poses in the context of existing binding pockets. Include also the reference structure in the figure.

Line 166: “To evaluate the ligand positioning within the 166

binding pockets, phytochemicals were subjected to docking against DPP-4 (Fig. 1).” How did you define the docking volume?

Line 167: “Among 167 the compounds, isorhamnetin, DTXSID90724586, and CHEMBL3446108 were selected 168 due to their inhibitory activity, docking scores, and interactions with DPP-4.” What inhibitory activity? Are these reference compounds ? If theses are docking hits, elaborate in detail how they are selected amongst docking hit-list.

Table 1: What binding affinity? Or is that just a docking score ?

Figure 1: Left 3d structures ae not informative: incorporate interacting residues and indicate where this is in the context of reference binding pockets. Include also the binding of reference compound.

Line 229: “Both substance molecules sat- 229 isfied Lipinski’s ‘Rule of Five,’ indicating they possess an advantageous balance between 230 lipid and water solubility.” What?

Chapter ADMET tox. How was this predicted?

Line 224: “he bioactivity score of these compounds was predicted” Elaborate a bit more on this step? What models were employed?

Chaper 2.4: When You introduce a short description for the first time “n has the lowest value at -1143.775 Eh” it should be described.

Do You expect these compounds to interact covalently? That is why you conducted this study?

Line 297: “The 296

differences in electronic energy suggested variations in the stability and bonding interac- 297

tions of these compounds” and “Higher energy values indicated more reactive compounds.” What? If You want to talk of reactvity for example, this should be elaborated, this kind of general statement is not suitable at this position.

Line 317: “Moreover, all three compounds diverged from the control in terms of 317 electronegativity and electrophilicity indicating differences in their chemical reactivity 318 and potential interactions with other molecules.” So you conclude all the molecules are different from this QM study? Do any of your results correlate with interaction potential You propose for them?

Why did You study HOMO-LUMO, is there a reactivity context with the target? Which chemism do You expect, indicate via a scheme.

Chapter 2.5. Indicate RMSF regions that correlate with Your binding site.

Do You expect SASA differences of the whole systems? Especially if You analyze the binding of your compounds.

Analysis of the bonds during the MD is maybe the most important observation for the point You want to make. Please do this analysis and provide schemes.

Line 407: “The MM-PBSA binding energies of the DPP-4-isorham- 407

netin, DPP-4-DTXSID90724586, DPP-4-CHEMBL3446108, and DPP-4-N7F complexes 408

were computed as -35.60 kcal/mol, 48.58 kcal/mol, 53.62 kcal/mol, and 37.57 kcal/mol, re- 409

spectively, it is indicated their robust binding to the protein.” So these are thermal MM-PBSA? Calculated along the trajectory and averaged? Please indicate this detail info.

Line 409: “s -35.60 kcal/mol, 48.58 kcal/mol, 53.62 kcal/mol, and 37.57 kcal/mol, re- 409

spectively, it is indicated their robust binding to the protein” From -35 to +53 is robus binding? Or You mean only for some? Elaborate.

Line 412: “The quantitative 412 MM-PBSA analysis validated the binding affinity that was observed in the protein-ligand 413 docking simulations.” Yes? Elaborate.

Line 427: “GFE” What is it? Introduce for the first time…. How is the GFE calculated herein? What do You observe from FEL that is relevant for DPPL4 inhibitor design? How does it correlate with previous studies?

Chapter 2.7. “Probability Density Function” Elaborate what was done here in more detail.

Line 443: “The proportions of eigenvector contributions to the total conformational variance were 443 determined to be 45.32% for isorhamnetin, 23.35% for DTXSID90724586, 24.66% for 444 CHEMBL3446108, and 28.26% for N7F, respectively (Fig. 5)” Do You observe herein specific conformational states of the protein. Elaborate how this is relevant for Your inhibitor design and your identified inhibitors.

Comments on the Quality of English Language

Minor edits are needed.

Author Response

Dear Reviewer: We would like to thank you for your time and effort in reviewing our manuscript, and valuable suggestions and corrections to improve the quality of our manuscript. We have carefully addressed all of the issues raised in the comments and incorporated in the text. The following is a point-by-point response to the reviewer's comments. We have marked all revised items in red color in the text according to the editors’ suggestions.

General Comments:

Manuscript “pharmaceutics-2889854” entitled “Characterization of Plant Derived Natural Inhibitors for Dipeptidyl Peptidase-4 as Potential Antidiabetic Agents: A Computational Study” describes an in-silico drug design study on DPP-4 target but lacks critical data and conclusions for the reader. I recommend major revision to amend the text:

Author Response: We thank the reviewer for careful reading and comments of the manuscript. 

Other comments

  1. Abstract: Line 32-44 – This info is not suitable for the abstract. Keep the abstract short and concise to include results only.

Response: We appreciate the reviewer's valuable comments and have already improved the abstract of our revised manuscript.

  1. Line 110-111; please provide the reference.

Response: Thank the reviewer for careful reading, and we have added the reference in the revised manuscript.

  1. Line 115-116; “Using computational approaches, this study identified phytochemicals that exhibit favorable interactions and binding affinities with the active site of DPP-4.” This study identified the compounds that have potential to bind to DPP-4. Rephrase!

Response: We thank the reviewer for careful reading and comments, and we have rephrased the sentence in the revised manuscript.

  1. Line 132: For example, Lipinski is not a filter to access drug-likeness. Each filter You describe has a specific purpose. Do You think phytochemical adhere to all Your employed filters in general, is it sensible to apply Lipinski filter at the beginning phytochemical design stage?

Response: We appreciate the reviewers and with agree the comments. Actually, phytochemicals may not consistently adhere to all filters, including Lipinski's, due to their diverse chemical nature. Applying the Lipinski filter at the early phytochemical design stage might not be universally sensible, as plant-derived compounds often deviate from conventional drug-like properties. Specific considerations for phytochemical structures are crucial in designing effective bioactive compounds. We have improved our understanding.

  1. Line 138: “After applying this rigorous filtration process, the final set of phytochemicals consisted of only 23.5% of” 23 % of What?

Please include the chapter: Library preparation where You describe in detail how You prepared the libraray and how many compounds are in there. Then You describe which compounds are filtered off in each step and supply the complete library in the supp info.

Response: Thanks for the careful reading and valuable suggestions to improve the manuscript. After applying this rigorous filtration process, the final set of phytochemicals consisted of only 23.5% of the total compound. We have corrected in the text.

We included the library preparation part following the instructions. Moreover, we have described the library of a complete list of phytochemicals with their CID obtained from 81 medicinal plants in this study.

  1. Line 143-165: Prepare a figure where you indicate these sites on the target and display your poses in this context so the reader can evaluate the docked poses in the context of existing binding pockets. Include also the reference structure in the figure.

Response: We have added the figure as supplementary fig. 1 following the instructions in the revised manuscript. We hope its help to the reader for easily understanding.

  1. Line 166: “To evaluate the ligand positioning within the binding pockets, phytochemicals were subjected to docking against DPP-4 (Fig. 1).” How did you define the docking volume?

Response: We thank the reviewer for the question. Docking volume defines to the three-dimensional space within which molecular docking simulations are performed to predict the binding orientations and conformations of ligands within the binding pocket of a target protein. We performed site-specific docking, where the binding site of the target protein was subjected to docking against DPP-4.

  1. Line 167: “Among the compounds, isorhamnetin, DTXSID90724586, and CHEMBL3446108 were selected due to their inhibitory activity, docking scores, and interactions with DPP-4.” What inhibitory activity? Are these reference compounds? If these are docking hits, elaborate in detail how they are selected amongst docking hit-list.

Response: We would like to clarify that we selected the top three compounds based on their docking scores and interaction types with the target protein. The mention of inhibitory activity was an error made during the writing process, and we have since removed it from the revised version of the manuscript.

  1. Table 1: What binding affinity? Or is that just a docking score?

Response: We appreciate the reviewer's careful reading and comments. We have corrected the docking scores in the revised manuscript.

  1. Figure 1: Left 3d structures are not informative: incorporate interacting residues and indicate where this is in the context of reference binding pockets. Include also the binding of reference compound.

Response: We have added the interacting residues in the 3d structure figure. I hope its helps to readers for easy understanding.

  1. Line 229: “Both substance molecules satisfied Lipinski’s ‘Rule of Five,’ indicating they possess an advantageous balance between lipid and water solubility.” What?

Response: The selected molecules satisfied Lipinski’s ‘Rule of Five,’ indicating they are considered as a lead compound. We have corrected the sentence in the revised manuscript.

  1. Chapter ADMET tox. How was this predicted?

Response: ProTox-II server used for toxicity prediction. It is integrate molecular similarity, pharmacophores, fragment propensities, and machine learning models to forecast diverse toxicity outcomes, including acute toxicity, hepatotoxicity, cytotoxicity, carcinogenicity, mutagenicity, immunotoxicity, adverse outcomes pathways (Tox21), and toxicity targets. The predictive models utilize data from both in vitro experiments, such as Tox21 assays, Ames bacterial mutation assays, hepG2 cytotoxicity assays, and immunotoxicity assays, as well as in vivo scenarios, including carcinogenicity and hepatotoxicity. These models have undergone validation using independent external datasets, demonstrating robust and effective performance (https://doi.org/10.1093/nar/gky318).

SwissADME and pKCSM are freely accessible web tools. It was used to predicted pharmacokinetics properties of small molecules. SwissADME are works based on a combination of established computational models and algorithms, including quantitative structure-activity relationship (QSAR) models and empirical rules derived from experimental data. These models use molecular descriptors and statistical correlations to make predictions about the pharmacokinetic properties of small molecules (https://doi.org/10.1038/srep42717).

  1. Line 244: “The bioactivity score of these compounds was predicted” Elaborate a bit more on this step? What models were employed?

Response: Molinspiration's Bioavailability Suite uses proprietary methods, including chemical descriptors and machine learning/QSAR modeling, to predict bioactivity scores based on molecular structures. The trained model estimates the likelihood of a molecule exhibiting specific biological activity. We have added the elaborate steps described in the revised manuscript.

  1. Chapter 2.4: When You introduce a short description for the first time “n has the lowest value at -1143.775 Eh” it should be described.

Response: We thank the reviewer for careful reading and comments to improve our manuscript. We have described the electronic energy (Eh) in the revised manuscript.

  1. Do You expect these compounds to interact covalently? That is why you conducted this study?

Response: We thank the reviewer for valuable comments. Yes, we have expected these compounds to interact covalently. That is why we conducted this study.

  1. Line 297: “The differences in electronic energy suggested variations in the stability and bonding interactions of these compounds” and “Higher energy values indicated more reactive compounds.” What? If You want to talk of reactivity for example, this should be elaborated, this kind of general statement is not suitable at this position.

Response: This is a general statement we have added this other place of this part in the revised manuscript. We hope this is suitable now.

  1. Line 317: “Moreover, all three compounds diverged from the control in terms of electronegativity and electrophilicity indicating differences in their chemical reactivity and potential interactions with other molecules.” So you conclude all the molecules are different from this QM study? Do any of your results correlate with interaction potential You propose for them?

Response: We thank the reviewer for careful reading and valuable comments. This sentence is a writing mistake. It actually “Moreover, all three compounds diverged from the control in terms of electronegativity and electrophilicity indicating differences in their chemical reactivity and potential interactions with reference molecule(N7F).” We have corrected in the revised manuscripts.

  1. Why did You study HOMO-LUMO, is there a reactivity context with the target? Which chemism do You expect, indicate via a scheme.

Response: HOMO-LUMO study was performed for describing the reactivity of molecular orbitals in organic molecules. The importance of the reactivity of DPP4 target is laid out in previous studies like in doi: 10.2147/DDDT.S86529. Molecules with a small or no HOMO–LUMO are preferred for activity. This chemism is studied for the potential inhibitors. We have added to the revised manuscript in the Frontier Molecular Orbital Analysis section.

  1. Chapter 2.5. Indicate RMSF regions that correlate with Your binding site.

Response: We have added the description in the revised manuscript according to the reviewer comment.

  1. Do You expect SASA differences of the whole systems? Especially if You analyze the binding of your compounds.

Response: From our understanding, we expect SASA differences of the system. When a compound binds to a biological target, changes in SASA reflect alterations in molecular conformation, interactions, and solvent accessibility that occur upon binding.

  1. Analysis of the bonds during the MD is may be the most important observation for the point You want to make. Please do this analysis and provide schemes.

Response: We appreciate for the very professional suggestion for the betterment of the manuscript. However, it is a limitation at this point in our manuscript which we could not provide.

  1. Line 407: “The MM-PBSA binding energies of the DPP-4-isorhamnetin, DPP-4-DTXSID90724586, DPP-4-CHEMBL3446108, and DPP-4-N7F complexes were computed as -35.60 kcal/mol, 48.58 kcal/mol, 53.62 kcal/mol, and 37.57 kcal/mol, respectively, it is indicated their robust binding to the protein.” So, these are thermal MM-PBSA? Calculated along the trajectory and averaged? Please indicate this detail info.

Response: The reported binding energies are the thermal MM-PBSA. The trajectories are given in the Fig 3. Computed values of -35.60 kcal/mol, 48.58 kcal/mol, 53.62 kcal/mol, and 37.57 kcal/mol were the average from 0 ns to 100 ns simulation runtime. We have included the texts in the revised manuscript.

  1. Line 409: “s -35.60 kcal/mol, 48.58 kcal/mol, 53.62 kcal/mol, and 37.57 kcal/mol, respectively, it is indicated their robust binding to the protein” From -35 to +53 is robust binding? Or You mean only for some? Elaborate.

Response: Actually, we mean only for some. As far as we know, higher positive and negative values of MM-PBSA analysis indicate more strong binding. If energies are more than -35 to +53 it indicates more strong binding of the complex.

  1. Line 412: “The quantitative MM-PBSA analysis validated the binding affinity that was observed in the protein-ligand docking simulations.” Yes? Elaborate.

Response: We thank the reviewer for careful reading and comments. The binding affinities predicted by molecular docking were validated through MMPBSA analysis of MD simulation. MMPBSA analysis complements molecular docking by providing a more detailed and accurate estimation of binding free energy and affinity, incorporating solvation effects, handling conformational flexibility, and validating the results obtained from docking simulations. We corrected the text in the revised manuscript.

  1. Line 427: “GFE” What is it? Introduce for the first time…. How is the GFE calculated herein? What do You observe from FEL that is relevant for DPPL4 inhibitor design? How does it correlate with previous studies?

Response:  GFE (Gibbs Free Energy) is often used to predict the stability and energetics of protein-ligand interactions, such as the binding of a drug molecule to its target protein. The GFE of a protein-ligand complex is computed by decomposing the total free energy into contributions from molecular mechanics energy and solvation energy. Energy minima, high energy barriers, overall shape, and depth of the FEL reflected the binding affinity of the inhibitor for its target. These parameters are observed from the FEL plot. These results were also in line with previous studies like doi:10.3389/fmolb.2022.1024764, doi: 10.1080/07391102.2023.2291831. Some of this text we added the revised manuscript.

  1. Chapter 2.7. “Probability Density Function” Elaborate what was done here in more detail.

Response: The PDF provides information about the distribution of structural states of a biomolecular system. It indicates the probability density of observing specific combinations of radius of gyration (rg) and root-mean-square deviation (rmsd) values within the system. The colorbar represents the intensity or probability density of observing specific combinations of rg and rmsd values. A good PDF represented a well-defined, smooth distribution with clear peaks of high density, covering the relevant range of rg and rmsd. We have elaborated the text in the revised manuscript.

  1. Line 443: “The proportions of eigenvector contributions to the total conformational variance were determined to be 45.32% for isorhamnetin, 23.35% for DTXSID90724586, 24.66% for CHEMBL3446108, and 28.26% for N7F, respectively (Fig. 5)” Do You observe herein specific conformational states of the protein. Elaborate how this is relevant for Your inhibitor design and your identified inhibitors.

Response: We thank the reviewer for careful reading and comments. Yes, each eigenvector represented a principal component of conformational mode and its contribution to the total variance indicated the importance of that mode in describing the overall dynamics of the system. PCA helped in characterizing the flexibility and dynamics of the protein-ligand complex by quantifying the extent to which different regions of the protein move or fluctuate during the simulation.

Round 2

Reviewer 1 Report

Comments and Suggestions for Authors

The manuscript was corrected according to the suggestions, therefore, I recommend its publication.